# A Visualization Analysis of Crisis and Risk Communication Research Using CiteSpace

**DOI:** 10.3390/ijerph19052923

**Published:** 2022-03-02

**Authors:** ShaoPeng Che, Pim Kamphuis, Shunan Zhang, Xiangying Zhao, Jang Hyun Kim

**Affiliations:** 1Department of Human-Artificial Intelligence Interaction, Sungkyunkwan University, Seoul 03063, Korea; cheshao@g.skku.edu (S.C.); zxy94@g.skku.edu (X.Z.); 2Department of Interaction Science, Sungkyunkwan University, Seoul 03063, Korea; p.i.m@skku.edu (P.K.); 970205@g.skku.edu (S.Z.)

**Keywords:** visualization, crisis and risk communication, CiteSpace, co-authorship analysis, co-citation analysis, turning points analysis

## Abstract

This study aims to understand the research status and development trend of crisis and risk communication research (CRCR) through a visual analysis in CiteSpace, thereby providing a more comprehensive perspective for future research agenda. First, we retrieved published papers from Web of Science (1986–2020) and Scopus (1979–2020) with a title search. Subsequently, we analyzed the main research strengths and main topics of CRCR from two dimensions: co-authorship network and co-citation network. We conducted an in-depth co-citation network analysis from four perspectives: cluster analysis, high co-cited literature analysis, burst analysis, and turning points analysis. These results revealed the main research topics in the CRCR field, the most eye-catching research literature, the emerging research hotspots in each period, and the turning points of the overall development. Finally, we suggested further research directions for future avenues.

## 1. Introduction

Since the beginning of the COVID-19 outbreak in December 2019, human societies have been facing a crisis caused by variants of the virus, such as Delta and Omicron [1,2]. Rapid vaccine development and communication strategies are challenges in the face of major public health emergencies [3]. The government and the media should use risk communication strategies to warn citizens about a potential epidemic development and flexibly use crisis communication to address the outbreak risks [4]. The repeated alternation of risks and crises poses great challenges to scholars and practitioners.

Crisis communication and risk communication are essential aspects and approaches of risk management and emergency management, respectively [5]. However, due to the confusion and overlap in their definitions, their reference and applications have become very complicated [6]. The fundamental reason is that although the boundary between these two concepts is clear in definition, the operable boundary in practical applications is fuzzy and mutually transformative in nature. For example, (a) when the risk communication fails, the potential threat will be directly transformed into a crisis. In addition, the subsequent communication will also become crisis communication [7]. Similarly, follow-up risk communication is still needed to prevent a second crisis [8]. Moreover, even in crisis communication, there are potential risks. (b) Both work to reduce the likelihood or extent of harm. (c) Finally, communication is the primary means of both. Thus, instead of attempting to separate the two concepts, they should be integrated. Therefore, with the emergence of the Crisis and Emergency Risk Communication (CERC) model [9], academic research on crisis and risk communication has entered a new stage.

Crisis and risk communication research (CRCR) have rapidly developed due to the importance of crisis and risk in many fields: social sciences, medicine, business, environmental science, arts and humanities, engineering, economics, econometrics and finance, computer science, medicine, management and accounting, and so on. However, few studies have systematically investigated CRCR’s comprehensive intellectual landscape.

Publications contain citations and references, which identify the sources used, demonstrate the depth of the research, put work into context, and acknowledge other scholars’ work. While there seems to be a lack of unifying theories on citations in academic works, Merton [10] argued from a normative perspective that citations are institutional forms to recognize and reward scientists’ work. Cronin [11] criticized the validity of citations’ use as a proxy for recognition and the quality of the cited work. However, he suggested that they can be utilized to shape a holistic understanding of the scholarly communication system [12]. From a social constructionism perspective, referencing in papers exposes connections between established theories and observations, resulting in the construction of consilience networks. In this paper, however, we use citations to determine the interconnected channels of scientific information and construct bibliometric networks.

Among the methods employed for the bibliometrics network construction, scientific metrology is one of the most effective to describe a specific academic field comprehensively and quickly [13]. It involves collecting all of the literature in a particular field within a specified period and constructing a review of the field corresponding to different data elements in the literature. Scientific metrology is a quantitative evaluation method for scientific activities based on bibliometrics and scientific statistics. Its evaluation process has strong objectivity, authority, simplicity, and repeatability. Objectivity means that scientific measurement is not affected by the subjective relationship of the “peer network” linked by the complex relationship between the academic and the social. Authority means that the scientific measurement method is based on Science Citation Index (SCI), Social Sciences Citation Index (SSCI), and other internationally famous literature retrieval tools, with strict screening standards and procedures. Finally, simplicity means that scientific measurement includes many accepted algorithms, and relevant indicators can reflect the “influence” of most papers.

As of 2021, few efforts have been made regarding “crisis communication” or “risk communication.” Among them, the most representative are by Goerlandt et al. [14], Dong et al. [15], and Deng [16]. In terms of risk communication, Goerlandt et al. [14] analyzed the field of risk communication using CiteSpace and identified geographic and national trends, patterns in scientific categories, and representative journals. Dong et al. [15] compared China and the United States in regard to CRCR for four dimensions—individuals, institutions, countries, and keywords. In terms of crisis communication, Deng [16] used CiteSpace to analyze 4116 crisis communication papers published on Web Of Science (WOS) from 2010 to 2020. The author identifies the most influential authors, institutions, theories, papers, hot topics, and future trends in crisis communication research. However, these works are not comprehensive. For example, these three studies used only one database and utilized incomplete search terms (although one search term can describe the field more accurately, the overlap between crisis communication and risk communication in the development process was ignored). Additionally, Goerlandt et al.’s [14] study focused only on the national level and lacked consideration of the role of individuals and institutions, such as author cooperation analysis, author co-citation analysis, institutional cooperation analysis, and journal co-citation analysis. Similarly, Dong et al. [15] studied only a small number of items in the comparative analysis and did not report whether the literature data in WOS excluded Chinese data. Deng [16] was unable to propose a research agenda based on the research results to suggest directions for future research. Moreover, no work has combined risk communication and crisis communication as search terms to explore the development context between the two. Based on previous work, this study proposes the following objectives:To identify the current state of cooperation at the national, institutional, and individual levels in the CRCR area as well as the most productive countries, institutions, and individuals;To identify highly regarded journals, literature and authors in the CRCR field;To discover major research topics in the CRCR field;To explore the turning points in the evolution of CRCR;To determine CRCR’s future research agenda.

Therefore, this study expands the scope of the literature by adding additional sources, search terms (risk communication and crisis communication), and time span. This study was conducted as follows:(1)The co-authorship analysis was conducted at three levels: country, institution, and author.(2)The co-citation analysis was conducted at three levels: journal, literature, and author.(3)The literature co-citation analysis was specifically expressed as cluster analysis, high-co-cited literature analysis, burst analysis, and turning points analysis.

This paper is organized as follows. In the first section, we introduce the research background, review the previous research, and propose the purpose of this research. The second section describes the structure of the methodology. The third section presents the results and corresponding discussion. Finally, the fourth, fifth, and sixth sections offer conclusions, suggestions for future research agendas, and limitations, respectively.

## 2. Methods

The method flow chart, shown in Figure 1, reports that this study comprises three steps: data collecting, data processing, and data analysis.

### 2.1. Data Collection

WOS and Scopus are considered to be the largest citation databases available as of 2021, containing information on literature in various disciplines such as medicine, science and engineering, the humanities, and art [17]. Additionally, Scopus offers a wider variety of information on the literature in the humanities field and a deeper history of publications dating back to the beginning of the 20th century. Therefore, this paper is based on merged citation data from both databases.

Specifically, this study employed the search terms “risk communication” or “crisis communication”. Moreover, to ensure that non-CRCR-centric articles do not appear among the search results, this research set the search scope as “TITLE.” Therefore, the following search queries were used for data retrieval:

Scopus: (TITLE ({risk communication}) OR TITLE ({crisis communication}) AND PUB YEAR > 1978 AND PUB YEAR < 2021.

WOS: “risk communication” (Title) or “crisis communication” (Title), Timespan: 1 January 1986 to 31 December 2020.

Ultimately, 4164 available results were retrieved in total: 2418 papers from Scopus and 1746 papers from WOS.

### 2.2. Data Analysis Tool

The use of bibliometric maps to represent how different types of objects of study (authors, papers, journals, organizations, etc.) are related to one another is considered a useful way to help its visualization and comprehension. Many software programs are currently used to perform bibliometric mapping: CiteSpace, VOSviewer, CitNetExplorer, SCI2, Sci2Tool, Pajek, and Gephi, among others. However, each software program has its advantages and disadvantages.

As Figure 2 shows, different software focuses on different areas of analysis [18,19]. Although some software such as Pajek and Gephi are more capable in network analysis and network visualization, respectively, CiteSpace overwhelms others in terms of comparative comprehensiveness. It covers most functions required by scientific research, such as turning point, dual-map overlay, and timeline.

CiteSpace is a citation visualization analysis software tool that focuses on analyzing the potential knowledge contained in scientific analysis, and has gradually developed against the background of scientometrics and data visualization. CiteSpace can make maximum use of the information contained in the literature to conduct a structured and timeline analysis of past research in the field, such as using author, institution, and country information to describe authors, institutions, and countries that have made significant contributions to the area over previous decades. CiteSpace can also use title, abstract, and keywords to predict the field’s future development. CiteSpace’s comprehensiveness is its most significant advantage, so we chose this software package as our research tool in this study.

### 2.3. Data Processing

#### 2.3.1. Data Format Conversion

Due to the difference in variables in the output files from Scopus and WOS, the datasets had to be converted for match and merge. After conversion, both datasets were merged into a single text file. The composition of the combined dataset is shown in Table 1.

#### 2.3.2. Data Deduplication

We used CiteSpace to remove duplicated data. During this process, 405 duplicate records were deleted; 3271 unique records remained in our revised dataset.

#### 2.3.3. Yearly Outputs

After cleaning and converting the dataset, a quantity chart of CRCR publications per year was acquired, as shown in Figure 3.

As shown in Figure 3, we found that the year 2020 (38.08%) had the highest growth rate (over 100 papers), possibly because the COVID-19 outbreak brought more interested researchers into the CRCR field.

#### 2.3.4. Software Parameter Settings

After data processing, we imported the dataset into CiteSpace for exploration and analysis. The specific parameter settings used in CiteSpace were as follows:

Time slicing includes “Time” and “Years Per Slice.” “Time” specifies the year range of publication. The starting point of this study is January 1979, and the ending point is December 2020. “Years Per Slice” divides data time zone in the unit of year, and one year is this study’s set value.

The map made by CiteSpace does not use the original co-occurrence matrix; it uses algorithms to normalize the matrix based on the original matrix and then uses the new matrix for network visualization. These algorithms include the Jaccard index, Cosine similarity measure, Dice coefficient, and Pointwise mutual informationConfiguring CiteSpace to display information about nodes whose frequency exceeds the target frequency.

The term *density* describes the strength or tightness of connections between nodes in a network and is calculated by dividing the “actual number of relationships” in the network by the “theoretical maximum number of relationships.” The more connections between nodes, the greater the density of the network is, and the greater the network’s density, the greater the influence of the network between nodes is. The values of density range from 0 to 1. When all nodes in the network are connected, the density equals one, and vice versa.

The *silhouette* value is a measure of network homogeneity. A silhouette value above 0.5 is considered reasonable, and one of 0.7 has a high confidence.

*Modularity* is the evaluation index of network modularization and is represented by the letter Q. Q value’s range is between 0 and 1. Exceeding 0.3 indicates that the resulting network community structure is significant.

*Betweenness centrality* is an indicator of a node’s importance based on the number of shortest paths that pass through it. CiteSpace employs this metric to discover and assess the impact of the literature on the network. When the betweenness value of a node exceeds 0.1, CiteSpace uses a purple circle to highlight the node.

### 2.4. Data Analysis

#### 2.4.1. Co-Authorship Analysis

Co-authorship is an association between two or more researchers who collaborate to report their research findings on a specific topic. This term can be thought of as reflecting social networks encompassing researchers that reflect collaboration among them. It is clear that the primary use of co-authorship networks is to investigate the structure and evolution of scientific collaboration [20,21]. Glänzel and Schubert [22] first explored scientific cooperation at the level of individuals and countries. However, because the intermediary role of institutions was ignored, this approach resulted in a gap in the research on cooperation between individuals and countries. When more boundaries are crossed, the collaboration becomes more productive and diverse [23], and the co-authorship becomes more likely. In contrast, Chen and Liu’s [24] analysis of scientific cooperation in high-speed rail research is more comprehensive as it explores productive individuals, institutions, and countries. Based on previous research, our study also included an analysis of the scientific cooperation network in CRCR from micro, meso, and macro perspectives.

The co-authorship analysis consisted of country co-authorship, institution co-authorship, and author co-authorship analyses.

#### 2.4.2. Co-Citation Analysis

When two journals, pieces of literature, or authors appear simultaneously in the references of a paper, they are said to have co-citation relationships [25]. If the co-citation frequency is high, the academic relationship between the two is close. This study analyzes co-cited journals, co-cited papers, and co-cited authors.

The co-citation analysis consisted of journal co-citation, literature co-citation, and author co-citation analyses. Among these, literature co-citation was examined using cluster analysis, high co-cited literature analysis, burst analysis, and turning points analysis.

## 3. Results and Discussion

Table 2 shows the parameters retrieved from the data analysis.

### 3.1. Co-Authorship Analysis on CRCR

#### 3.1.1. Country Co-Authorship Analysis

Figure 4 depicts CRCR’s national cooperation network. Each node represents a country, and its size represents the number of papers. The distance and thickness of the edges between nodes represent the degree of cooperation among countries. The node’s purple border represents the node’s betweenness centrality, or its frequency acting as a bridge in the shortest path between the other two nodes [26]. As a result, the greater the centrality of a node’s betweenness, the thicker the purple ring, or the higher the frequency at which it acts as an “intermediary.” In addition, the circles inside the nodes represent the number of papers published in that country in a given year, with different colors representing different years. The deeper the ring’s color, the earlier the paper was published, and the lighter the ring, the closer it is to 2020. The specific parameters are shown in Table 2.

As shown in Figure 4, there were papers published in the US from 1979 to 2020, while those in China were predominantly published after 2010. Three nodes with high betweenness centrality, namely the US, England, and the Netherlands, play a crucial role in cooperation in CRCR. Among these, the US ranked first with evident advantages.

Table 3 shows the list of the top 10 countries in terms of the number of papers.

Overall, Europe lags behind the US in the number of papers, but the inclusion of five European countries in the Top 10 illustrates Europe’s advantage in the collaboration. North America and Asia both hold two positions in the top 10, while Oceania holds one.

#### 3.1.2. Institution Co-Authorship Analysis

Figure 5 shows the collaboration network between institutions in the CRCR domain. Using the clustering function, we can distinguish cooperative groups more intuitively [27]. The nodes in the diagram below represent institutions, and the size of the nodes indicates the number of papers published by the institution. The distance between nodes represents the degree to which two institutions cooperate. The closer the distance, the higher the cooperation frequency.

As shown in Figure 5, because the subnetworks in the cooperative network of institutions are not isolated from each other, we used the clustering function to observe the closeness of the relationships among institutions more clearly. CiteSpace shows that the modularity value (Q) is equal to 0.8977, which indicates that network community structure is significant. The weighted mean silhouette (S) is equal to 0.9997, which means that the network homogeneity has a high confidence. Ultimately, we found 13 subnetworks. The intensity of the subnetworks is marked in order, from 0 to 12, with 0 as the largest cluster and 12 as the smallest.

The core institution of subnetwork #0 is the University of Alabama. The numbers of papers of other nodes in the “0” subnetwork are all less than 15. However, they are still ranked first, indicating a high intensity of internal cooperation. Furthermore, subnetwork #1 is a subnetwork with Maastricht University and the University of Twente as the core, and the number of papers of other nodes in the subnetwork is fewer than 15. While #2 is the subnetwork with the University of Washington and Duke University as the core, subnetwork #3 is based on the core of Wayne State University, the University of Kentucky, and North Dakota State University. The centers of the networks represented by #4 are Cornell University and Columbia University, whereas the center of the network represented by #5 is the University of California, Rutgers University, and Harvard University. The number of papers from these three nodes are all in the top 10, but the frequency of cooperation is not high. Subnetwork #6 represents the network that contains four nodes with more than 15 papers: the University of Maryland, Virginia Commonwealth University, the University of Central Florida, and Louisiana State University. Sub-networks #7–#12 are provided in Figure 5.

Table 4 shows the list of the top 10 institutions in terms of the number of papers.

The network of institutional cooperation is different from the co-authorship network in that the latter has separated clusters. The network of institutional cooperation is a complete structure in which all institutions cooperate in part, forming a dense cooperative relationship. The top 10 institutions are of the same order of magnitude in terms of papers, but American universities dominate the top nine, showing strong research capabilities. Only the UK’s Cardiff University makes it into the top 10 list at the 10th place.

#### 3.1.3. Author Co-Authorship Analysis

Figure 6 shows the scientific collaboration network of authors in the CRCR field. The node represents the author, and the node size represents the number of papers. The larger the node, the more papers the author has published. Additionally, the degree of cooperation among authors is indicated by the thickness of the edge between nodes, with greater thickness indicating greater cooperation among authors.

We ensured that only the names of authors with more than five papers were shown. The color representing the year the publication was published in was coded as follows: the closer the edge is to purple, the closer the publication year is to 1979. The closer the edge is to yellow, the closer the publication year is to 2020.

As shown in Figure 6, we found a total of eight cooperative subnetworks and six independent authors. We selected the top four cooperation groups according to their cooperation intensity ranking. The first subnetwork to emerge was an early active three-person cooperative group comprising Caron Chess, K.L. Salomone, and B.J. Hance. The second subnetwork was a collaborative group with Adrian Edwards, Glyn Elwyn, C. Atwell, and Ian Russell as the prominent members (the number of papers is more than five). This group contains more than 10 authors and is the academic circle with the most significant cooperation intensity. They were most active between 2000 and 2010. The third subnetwork was an intermediate group of three with Matthew W. Seeger, Timothy L. Sellnow, and A. Schwarz. The final subnetwork was a recently emerged two-person group that included W.T. Coombs and S.J. Holladay.

As shown in Table 5, the betweenness centrality of all authors is 0, indicating that no author acts as a “bridge” in this field. Our in-depth analysis of the top 10 most prolific authors is shown below.

Table 5 shows the list of the top 10 authors in terms of the number of papers.

Adrian Edwards, a general practice professor in the population medicine department of Cardiff University, UK, is ranked first with 19 papers, and his main research content is related to medical risk communication. W. Timothy Coombs is ranked second with 16 papers. As a professor of liberal arts at Texas A&M University, his professional field is organizational communication. Ann Fisher, a senior scientist in the Department of Agricultural Economics and Rural Society at Pennsylvania State University, is ranked third and her main research field is human dimensions of integrated regional assessment. Caron Chess is an associate professor in the Department of Public Health Sciences at Rutgers University. She has published 13 papers in the fields of health communication, health literacy, health equity, and older adults. Professor Robert L. Heath, positioned in the Department of Communication at the University of Houston, has devoted himself to the areas of public relations, crisis communication, problem management, risk communication, and business-to-business communication with 13 articles. Glyn Elwyn has published seven papers in the field of medicine. Yan Jin has released 10 articles in the areas of crisis communication and strategic health risk communication. Ortwin Renn has put out 10 papers on risk governance. Matthew W. Seeger has offered nine papers in the area of risk communication and disasters. Finally, Timothy L. Sellnow contributed nine papers in the areas of bioterrorism and pre-crisis planning.

In CRCR, group cooperation is common; however, some authors continue to conduct research by themselves. Additionally, among the top 10 authors in CRCR, American authors made significantly more contributions than scholars of other nationalities; they were followed by British and German authors.

### 3.2. Co-Citation Analysis on CRCR

#### 3.2.1. Journal Co-Citation Analysis

The co-citation network diagram among journals in the CRCR field is shown in Figure 7. Each node corresponds to a journal. The higher the node, the more frequently the journal is cited. The distance between nodes represents the journal co-citation relationship. The greater the distance between two nodes, the greater the frequency with which the two journals co-cite each other.

As shown in Figure 7, the larger the node, the yellower the color, and the smaller the node, the more purple the color. We found that journals published after 2000 were cited more often, while earlier journals were cited less often. As shown in Table 6, Risk Analysis ranked first with 378 papers, followed by Science with 302 papers. Risk Analysis and Science are the two most-cited journals, and there is a high degree of overlap between them. This indicates that the two journals are often cited by other papers and have similarities in terms of research field. As the third most-cited journal, Public Relations Review partially overlaps with many smaller journals.

As the statistics in Figure 8 show, 10 SSCI and four SCIE journals are included in the top 10 most frequently cited journals distributed across the disciplines of social sciences (4), interdisciplinary (4), communication (3), medicine, general and internal (2), mathematics (2), psychology (1), information and library science (1), and business (1).

#### 3.2.2. Literature Co-Citation Analysis

Counting citations adds value to the excellent research work in a field, while higher co-citations play an essential role in reflecting the academic structure and tracking the evolution of a particular field [28,29]. Nodes represent co-cited references, as shown in Figure 9. The larger the node, the higher the frequency of citation [30]. The distance between nodes represents the frequency with which two pieces of literature are cited together. As a result, the closer the distance, the higher the co-citation frequency and the closer the research topic [24,31].

##### Cluster Analysis

As shown in Table 7, we obtained 12 clusters. According to the recommendation of Citespace, the most significant four clusters were reserved as our analysis objects. The largest cluster (#0) has 90 members and a silhouette value of 0.934. It is labeled as risk communication by the log-likelihood ratio (LLR). The most relevant citer for this cluster is “Parameters for crisis communication” [32].

Cluster #0 suggests that crisis communication has received significant attention in the last ten years, but much of the terminology is confusing. Coombs defined critical terms in crisis management and illustrated some of the key research on crisis communication [32]. Therefore, based on the standardization of communication parameters, many studies related to crisis communication or risk communication have been promoted, and these studies involve diverse fields. For example, the application of health risk communication in vehicular high power microwave system and deployment environment [33], the application of crisis communication in spokesperson crisis [34], the role of crisis communication in enterprises [35], and the preventive role of risk communication in bioterrorism, etc. [36].

Secondly, Coombs’ study also encouraged many researchers to focus on the development status of risk communication and crisis communication in their countries. For example, Magen [37] conducted a meta-analysis of crisis communication studies in Israel, examining crisis types, disciplines, and research priorities. Schwarz [38] reviewed crisis communication studies in Germany.

We should not ignore that although a vital research route of crisis communication is from case study to theoretical study, we find that in the distribution of experimental methods, cases and theories coexist. For example, Zhang and Vos [39] investigated the Facebook discussion event triggered by the disappearance of Malaysia Airlines Flight MH370 in 2014; Boys explored the crisis and risk communication issues in the case of sexual abuse by the Roman Catholic clergy; Gunawan et al. [40] studied the impact of crisis communication strategies and media reports on catering enterprise image.

Overall, the work of Coombs has been of great help to the risk and crisis communication practitioners and has promoted the use of relevant concepts and terminology by communication researchers in various countries and industries.

The second largest cluster (#1) has 82 members and a silhouette value of 0.947. It is labeled risk communication by the LLR. The most relevant citer to the cluster is “Social media engagement for crisis communication: a preliminary measurement model” [41].

Cluster #1 indicates that social media has increasingly become an essential part of crisis and risk communication, and the unique two-way communication model of social media has completely changed the one-way communication model of traditional media. Jiang and Luo [41] discussed crisis communication in social media and measured the use of social media in crisis management. This study integrated several popular theories in the field of crisis communication, such as attribution theory, image repair theory [42], SCCT [43], and SMCC [44]. The author proposed that the theoretical framework of crisis response should be centered on SCCT and SMCC. Addressing the public’s emotional response in crisis communication is the current and future focus.

Cluster #1 contains some bibliometric studies on crisis communication. For example, Harker and Saffer evaluated the development of crisis communication in sports [45], and Huang et al. discussed the forms and practices of crisis communication in Mainland China. The complex relationship between Chinese tradition and institutional background was elaborated, especially the political and media system related to the practice of crisis communication in China [46].

Pre-crisis communication is an important research topic in cluster #1. Generally speaking, the stages of crisis communication can be divided into three types [47]: pre-crisis, crisis, and post-crisis. However, there are also different classification methods. For example, Crisis Emergency Risk Communication (CERC) divides crises into five stages according to their life cycle: pre-crisis, initial, maintenance, resolution, and evaluation. Regardless of the classification method, we find that pre-crisis communication is always an independent stage and cannot be mixed with other stages. The pre-crisis stage is the risk stage, which is focused on discovering and identifying potential risks that may lead to a crisis. The pre-crisis strategy involves researching and gathering information about organization-specific crises and risks [48]. Then, a crisis management plan is developed, which includes making decisions ahead of time to determine when and where a crisis will occur and who will deal with specific aspects of the crisis. Finally, the organization should organize a rapid response crisis communication team in the pre-crisis phase, and all individuals who can help with the actual crisis communication response should receive training.

The third largest cluster (#3) has 47 members and a silhouette value of 0.943. It is labeled own risk by the LLR. The most relevant citer to the cluster is “Translating evidence-based information into effective risk communication: current challenges and opportunities” [49].

Cluster #3 shows that health risk communication is an important research topic. Although there are many quantitative and qualitative risk communication methods, there also are still many problems translating evidence into risk communication [49]. Cluster #3 contains many different health risk communication research directions including medical decision-making. For example, in many cases, it is difficult for patients to balance the advancement of technology with the size of side effects [50]. Patients cannot correctly understand medical probability issues, which pose significant challenges to the accurate perception of health risks and the ability to make medical decisions [51].

Another study is about the ethics of risk communication: doctors’ consultation mode plays an essential role in doctors’ clinical decision-making. Moreover, it has a direct influence on patients’ trust in doctors [52]. To address these issues, Dolan found that a combinational risk presentation was more likely to convey risk information to clinical decision-makers [53], and Ruland proposed informatics tools for shared decision making and risk communication to improve patient safety perceptions [54].

Health risk communication focuses on an early stage [55]. Health risk communication refers to information exchange on how to prevent, mitigate, and manage human health hazards, and it is a type of protection communication [56]. It is characterized by being multisubject and multiprocess, important aspects of crisis emergency management. In the case of public health events, the government formulates corresponding communication strategies for different subjects, the timely transmission of risk information among subjects, and the reduction of health risks caused by public health events. Health risk communication emphasizes the participation of multiple subjects whose purpose is not to reduce one subject’s anxiety and avoidance behavior but to build a multisubject cooperative group committed to solving the problem. Future research needs to assess the effectiveness of novel risk communication models based on patient and physician characteristics and identify appropriate models for translating evidence (quantitative or qualitative information).

The fourth largest cluster (#4) has 44 members and a silhouette value of 0.982. It is labeled social-mediated crisis communication (SMCC) research by the LLR. The most relevant citer to the cluster is “Crisis informatics in the context of social media crisis communication: theoretical models, taxonomy, and open issues” [57].

Cluster #4 refers to the crisis communication theory based on SMCC. Since 2010, social media has played an increasingly important role in crises [58]. As a result, an increasing number of relevant studies have been conducted, gradually forming more systematic theories, among which the SMCC model incorporating new media into crisis communication is prominent [59]. SMCC divides people into three types according to their differences in expressing social media messages during the crisis: influential social media creator, social media followers, and social media inactive. Communication media is divided into three categories: social media, traditional media, and offline word-of-mouth communication. In addition, to better present how the organization responds to the crisis through social media, traditional media, and offline word-of-mouth communication, SMCC analyzes an organization’s response to a crisis by examining its crisis origin, crisis type, infrastructure, message strategy, message form, and influential social media creators. In subsequent studies, several authors conducted in-depth studies on SMCC through empirical methods, and SMCC has gradually become a critical analytical framework for current crisis communication.

##### High Co-Cited Literature Analysis

Two of the top 10 most frequently cited documents are books, which are not included in our study. The other eight papers are selected for further analysis, as shown in Table 8.

**Table 8 ijerph-19-02923-t008:** Top eight co-cited journals for crisis and risk communication research.

Paper	Year	Times of Co-Citation	Method	Structure
Schultz et al. [28]	2011	29	Survey	3 (medium: newspaper, blog, twitter) × 3 (reaction: information, apology, sympathy)
Utz et al. [60]	2013	23	Survey	2 (crisis type: intentional, victim crisis) × 3 (media type: Twitter, Facebook, newspaper)
Jin et al. [61]	2014	15	Survey	3 (crisis information form) × 2 (crisis information source) × 2 (crisis origin)
Edwards et al. [31]	2002	13	Review	None
Coombs [62]	2015	12	Review	Crisis response strategies + situational crisis Communication theory
Coombs and Holladay [63]	2009	12	Experiment	2 (media: print, video news report) × 2 (response strategy: compensation, sympathy)
Coombs [64]	2007	11	Review	Attribution theory + situational crisis communication theory
Ott and Theunissen [65]	2015	11	Data analytics	Situational crisis communication theory

As shown in Table 8, Schultz et al. [28] surveyed 6814 employees to assess the impact of crisis communication strategies and media on organizational reputation, secondary crisis communication, and secondary crisis response. Their findings indicated that media type is more important than information. When compared to blogs and newspaper articles, crisis communication via Twitter results in fewer negative crisis responses.

Utz et al. [60] researched using the networked crisis communication model. The Fukushima Daiichi nuclear disaster was used to compare the effects of media and crisis type. The results indicated that the influence of media is more potent than that of the crisis type. Compared to the crisis communication of newspapers, crisis communication through social media can obtain a better reputation and reduce the secondary crisis response, such as the boycotting of companies.

With the public increasingly using social media for crisis discussion, crisis communication professionals must understand the key factors influencing the public’s consumption of crisis information via social media. As a result, Jin et al. [61] devised a 3 × 2 × 2 hybrid experiment based on the social-mediated crisis communication model that combined crisis information form, crisis information source, and crisis origin. They reported that the origin of the crisis has a significant impact on the form and source of information preferred by the public. This influences the public’s expectations of how organizations should respond to the crisis and the emotions people may experience when exposed to crisis information.

Edwards et al. [31] suggest that medical professionals should avoid framing information and encourage patients to make the right choices by converting raw data into information formats that are more conducive to discussion.

Based on previous research on crisis communication, Coombs [62] developed crisis response strategy guidance to help managers better communicate crises, including the expected results of the strategy, crisis response time, and factors affecting strategy effectiveness.

The difference in the effects of crisis communication conveyed via different media has always been topic of research interest. Coombs and Holladay [63] designed a 2 × 2 experiment with crisis response and media as impact factors. Crisis response included sympathy and compensation, and media included print and video. The experiment showed that the two influencing factors have similar results, and crisis managers can use the same strategy in both media channels.

The field of crisis and risk communication has long been dominated by case analysis, which is a type of nonsystematic data decision-making orientation. Coombs [64] reported that it is necessary to turn to empirical evidence and provide tested results to crisis managers to make evidence-based decisions. Attribution theory allows the integration of research results from different fields. Therefore, Coombs et al. [64] discussed the role of attribution theory in providing a comprehensive mechanism for crisis communication.

Inappropriate and risky communication strategies affect the reputation of enterprises. Ott and Theunissen [65] analyzed the social media crises experienced by Facebook, Greenpeace, Applebee’s, and Jetstar in 2010. They found that the participation strategy of risk communication affects the result, and different strategies are needed to contact users in different emotional states. Otherwise, the effect of risk communication will deteriorate. In addition, authenticity and transparency are key factors for success.

##### Burst Analysis

CiteSpace provides Burst detection to detect significant changes in citations during a period. For example, it is used to discover a subject’s decline or a keyword’s decline or rise.

In chronological order based on the outbreak, as shown in Figure 10, the top ranked item in terms of bursts is Edwards et al. [31], with bursts of 7.32. This is a research work on risk communication in the medical field, and the author considers whether the transformation of digital data into pictures in medical consultation can better convey risk information to patients. The second is Hearit [66], with bursts of 5.27. This is a book in behavioral science in which the author explores apologia and apologies as a communications strategy for corporations in crisis. The third is Lipkus [67], with bursts of 4.91. We find that the author developed this paper based on Edwards’ work through the burst year. The way risk information is presented in medical consultations continues to focus on research. The authors provide good practice on the extent to which health risks are expressed in numerical, verbal, and visual forms. The fourth is Coombs [64], with bursts of 5.79. This paper indicates that attribution theory has been formally introduced into crisis communication. The author guides the transformation of crisis communication research from case study to empirical study and discusses the role of attribution theory in providing a comprehensive mechanism for various crisis studies. The fifth is Coombs and Holladay [63], with bursts of 5.42. Post-crisis communication has gradually become a research hotspot. The author evaluates the impact of different coping strategies and media channels on respondents in crisis. The sixth is Coombs and Holladay [68], with bursts of 4.54. Apologies seem to have been overused as a communication strategy between 2007 and 2010. The authors compare apologies with more equivalent crisis strategies to determine whether apologies are the “best” strategy. The seventh is Schultz et al. [28], with bursts of 13.16. Since 2011, the popularity of social media has changed the traditional mechanism of crisis communication and the recipient’s reputation perception. Therefore, the author compares the influence of traditional and social media strategies on the recipient’s reputation perception through experiments. The eighth is Liu et al. [69], with bursts of 4.37. As a new communication mode, SMCC has not been an object of any empirical evaluation. Therefore, the author tested the SMCC model to explore the impact of the form and source of crisis information on public acceptance of crisis response strategies and public crisis emotions. The ninth is Utz et al. [60], with bursts of 9.67. Classical crisis communication theories ignore the role of media, so the author compares the relationship between media and crisis types. Finally, the tenth is Utz et al. [61], with bursts of 6.78. The authors also continue SMCC’s work by exploring the critical role of crisis sources in influencing the form and source of information that the publics prefer.

After further classification, we find that these ten papers can be divided into four types: risk communication (medicine), crisis communication (apology and attribution theory), crisis communication (medium), and SMCC. Among them, hot spots of risk communication in medical aspects concentrated from 2003 to 2012. Apology and attribution theories are focused upon from 2007 to 2013. With social media’s popularity, research on the relationship between crisis communication strategies and channels broke out from 2010 to 2018. Finally, until 2016, research based on SMCC has gradually occupied the mainstream academic platforms.

##### Turning Points Analysis

A paradigm is a broader phenomenon that corresponds to the clustering that occurs naturally in the scientific literature. The nodes that play a key role in paradigm shift are called turning points, which are bridges connecting different clusters.

As shown in Figure 9 and Table 9, after CiteSpace calculation, we found four turning points in total: Coombs and Holladay [63], Utz et al. [60], Seeger [70], and Falkheimer and Heide [71].

Figure 9 shows the development of CRCR research. The evolution of the network can be observed from right to left. Before 2006, the literature on CRCR focused on the analysis of own risk (#3), contingency theory (#6), skill development (#11), and risk communication research (#13). Until 2006, Seeger [70] had described ten best practices for effective crisis communication in a comprehensive analysis and taken these best practices as a theoretical method with a foundation to improve the effectiveness of crisis communication, especially in the context of large public management crises. However, Falkheimer and Heide [71] were not satisfied with the current situation of case-based research; thus, they conducted a critical analysis of research on crisis communication and cross-cultural public relations and developed different theoretical frameworks. This study also marks the formal shift of the center of crisis communication research from the case study method to theoretical framework guidance.

Coombs and Holladay [63] believed that because crisis communication was utterly dependent on news publications, people had little understanding of its channel effect, so they assessed the impact of different coping strategies and media channels on respondents in crisis. Coombs and Holladay’s [63] study on channels has illuminated new directions for crisis communication. Social media plays an important role in the negotiation and dynamics of crises in today’s society. Utz et al. [60] believed that classical crisis communication theories ignored the role of media, so they compared the influence of media and crisis type based on the network crisis communication model. The study also formally introduced social media into crisis communication theory, which, in turn, led to an explosion of research in several directions, for example, social media engagement (#1), social-mediated crisis communication research (#4), decision outcome (#5), and health crises (#9).

#### 3.2.3. Author Co-Citation Analysis

Author co-citation analysis can identify authors who investigate similar research topics as well as their research framework in this field by analyzing their papers [72]. The node represents the author, as shown in Figure 11. The greater the size of the node, the more the author is cited. The distance between nodes represents the frequency with which authors co-cite one another. The shorter the distance, the higher the frequency of co-citations and the more similar the research directions of the two authors.

As shown in Figure 11, we discovered that a large number of cited authors’ papers were published after 2010, and even more were published after 2015. The results can be divided into two groups. In the first group, Paul Slovic, and Fischhoff B, Covello VT, Renn O, Kasperson RE, and Sandman PM are the subnetworks surrounding nodes. For instance, Paul Slovic studies judgment and decision-making processes, focusing on decision-making under risk conditions, such as the influence of emotion on judgment and decision. In the second group, W. Timothy Coombs, Benoit WL, Seeger MW, Heath RL, Ulmer RR, Sellnow TL, and Yan Jin were subnetworks surrounding the nodes. W. Timothy Coombs’ primary research content is SCCT, a framework that provides academic help for strategy selection in coping with crises and uses theories to guide specific practices and operations.

Through the analysis, we find that the research fields within CRCR are primarily divided into two types: one is the relationship between cognition and risk, and the other is the risk management framework with SCCT as its core. The former focuses on theoretical research, while the latter focuses on practical operation.

Table 10 shows the list of the top co-cited authors in crisis and risk communication research.

## 4. Future Avenues for Research

Falkheimer and Heide [71], as essential contributors, led the research in the CRCR field from case study to theoretical framework guidance, which still offers important enlightenment for future research. Particularly through the COVID-19 situation, we found that urban lockdown was a possible government response to major health emergencies [73]. Social media has incomparable advantages in maintaining the normal operation of cities and the rapid and effective dissemination of government information [74]. Not only are Instagram and TikTok on the verge of replacing Facebook and Twitter [75], but social media messages are presented in a completely different way [76]. Photo sharing represented by Instagram and video sharing represented by TikTok have gradually become the primary approach for information dissemination. Therefore, we must consider whether the original theoretical framework can cover new forms of information presentation.

Coombs and Holladay [63] and Utz et al. [60] showed the influence of channels. Nonetheless, the classification of channels was relatively rough in early studies (of course, social media platforms were more singular at that time). Therefore, future research is necessary to study the impact of communication strategies on respondents in crisis based on different platforms. In addition, we believe that the presentation form of social media information has an expected differential impact on public health events, corporate crises, and environmental and energy crises.

## 5. Conclusions

To understand the evolution of CRCR, we conducted a comprehensive scientometric analysis of the CRCR field using co-author and co-citation analysis. In this study, we used CiteSpace as a research tool and WOS and Scopus papers published from 1978 to 2020 as research objects.

First, the co-authorship analysis conducted in this paper can be divided into three dimensions: country, institution, and individual. At the country level, the US, England, Germany, China, and the Netherlands are the five countries with the highest paper output. The US, England, and the Netherlands have made outstanding contributions to international cooperative research, and the US has a considerable advantage in terms of volume and field of cooperation. At the institutional level, the network centered on the University of Alabama has the highest cooperation intensity, forming a close cooperative relationship. The University of California, the University of Maryland, and Wayne State University are highly productive. At the individual level, a total of nine groups were identified, among which the group with Adrian Edwards as the core and Glyn Elwyn, C Atwell, and Ian Russell as the team members had the highest intensity. In addition, the most productive author was Adrian Edwards from Cardiff University in UK. W. Timothy Coombs from Texas A&M University in the US and Ann Fisher from Pennsylvania State University in the US ranked second and third, respectively.

Second, the co-citation analysis included three dimensions: journal, literature, and individual. At the journal level, Risk Analysis ranked first with 378 citations, and Science and Public Relations Review followed with 302 and 297 citations, respectively. Among the top ten most frequently cited journals, there were ten subcategories of SSCI journals and four subcategories of SCI journals. Humanities and social sciences occupied a dominant position, and the top four disciplines were social science (4), interdisciplinary (4), communication (3), and medicine, general and internal (2).

At the literature level, the cluster analysis showed four main themes in the CRCR field: risk communication (parameters), risk communication (social media), health risk communication, and social-mediated crisis communication research. Different topics contain many different research directions. For example, risk communication (parameters) includes the standardization of crisis communication, the development status of risk communication in different countries, and the coexistence of case studies and theoretical studies. Risk communication (social media) includes measuring the use of social media in crisis management, bibliometrics research of crisis communication, and pre-crisis communication. Health risk communication consists of the ethics of medical decision-making and risk communication. Finally, as the most popular model, social-mediated crisis communication research has become a unique topic.

The high co-cited literature analysis showed that Schultz et al. [28] were cited with the highest frequency (29 citations). According to the burst analysis calculation, we found that hot spots generated between 2000 and 2020 can be divided into four types: risk communication (medicine), crisis communication (apology and attribution theory), crisis communication (medium), and SMCC. Risk communication in medical aspects attracted great attention from 2003 to 2012. Apology and attribution theories were key research agenda from 2007 to 2013. From 2010 to 2018, studies on crisis communication strategies and channels were highlighted. Finally, research based on SMCC also increased its presence. The turning points analysis showed that the research context of CRCR moved through four stages: case analysis, theoretical framework, channel research, and social media. The US holds nine of the top ten spots at the individual level, with Paul Slovic from the University of Oregon ranked first with 404 total citations.

## 6. Limitations

First, although the keywords of risk communication and crisis communication selected in this study met the research purpose of this paper, they cannot form complete coverage of health communication. In addition, we did not include databases from other medical fields in this study.

Second, this study explored some correlation between risk communication and crisis communication in the development process; thus, the two search terms were combined. Third, future studies can perform a comparative analysis of risk communication and crisis communication to explore the differences in research fields between the two search terms.

Fourth, this study covered all of the years that included relevant literature, a relatively large time span; thus, it is necessary to carry out a nuanced analysis of the literature data of the past 5 and 10 years so that the results reflect more accurate information.

Finally, most studies, including ours, only examine developed countries. We may always tend to examine the top 10 countries, organizations, and authors, and developed countries are more likely to be in the top 10. Future research should focus on some of the countries we missed in our list.

## Figures and Tables

**Figure 1 ijerph-19-02923-f001:**
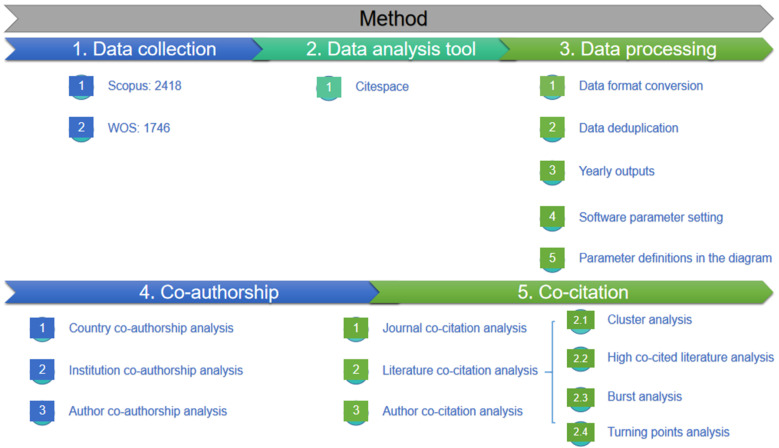
Research method flow chart.

**Figure 2 ijerph-19-02923-f002:**
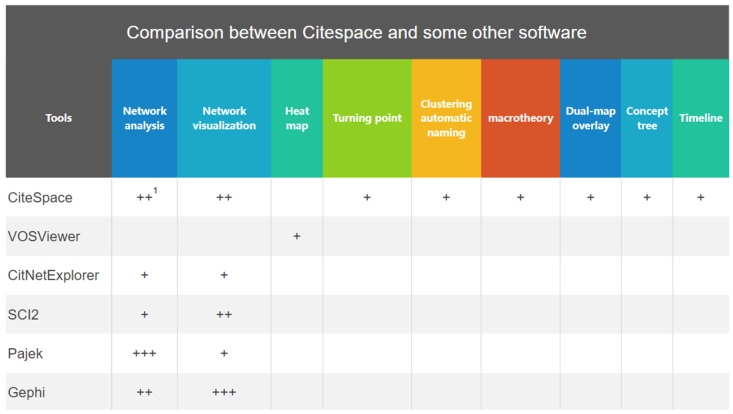
Comparison between CiteSpace and other software packages. ^1^ Note. +, ++, and +++ indicate that the software is weak, medium, and strong in the application field, respectively.

**Figure 3 ijerph-19-02923-f003:**
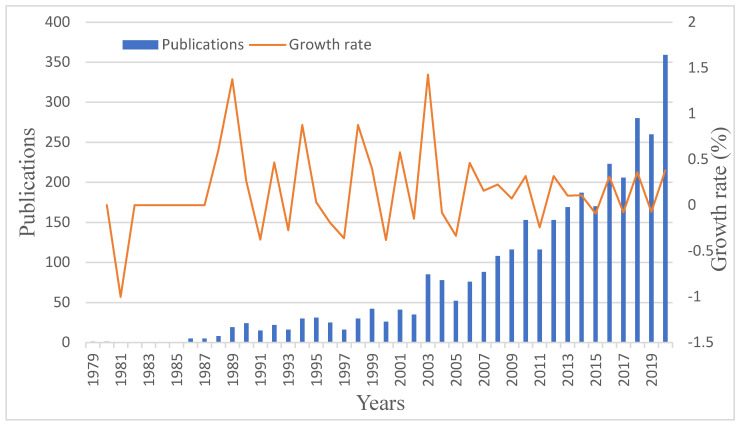
Yearly publication chart showing crisis and risk communication research between 1979 and 2020 (WOS and Scopus).

**Figure 4 ijerph-19-02923-f004:**
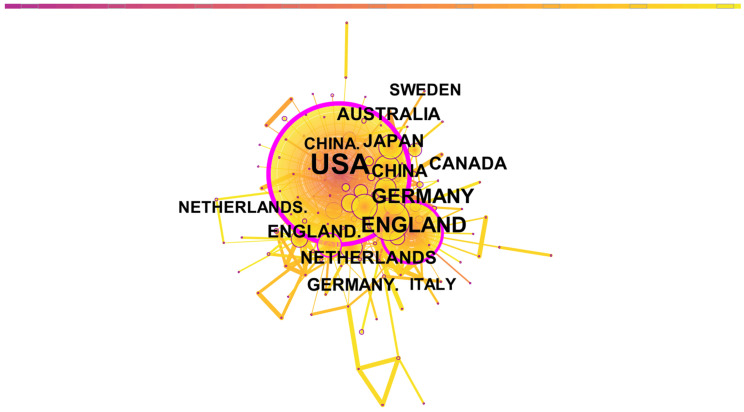
Map of the country co-authorship network in crisis and risk communication research.

**Figure 5 ijerph-19-02923-f005:**
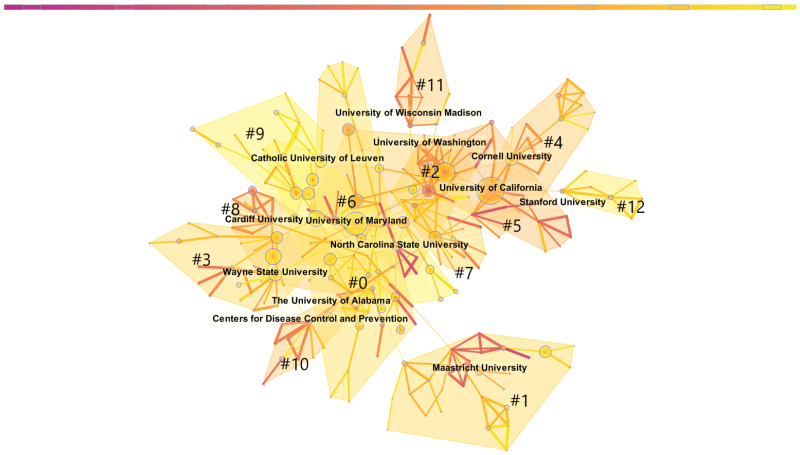
Institution co-authorship network of crisis and risk communication research.

**Figure 6 ijerph-19-02923-f006:**
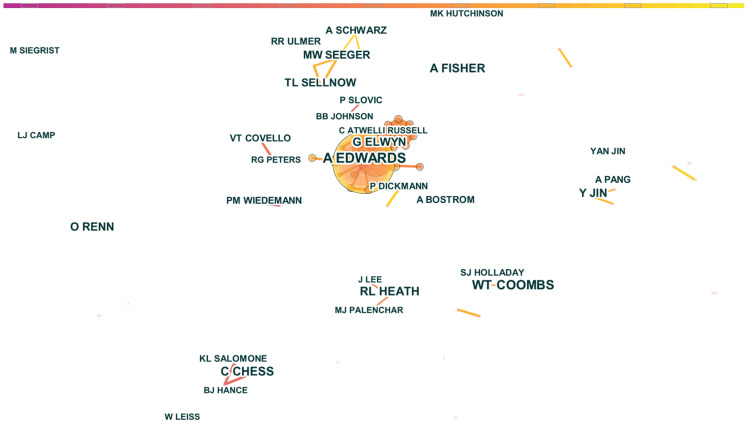
Map of the author co-authorship network of crisis and risk communication research.

**Figure 7 ijerph-19-02923-f007:**
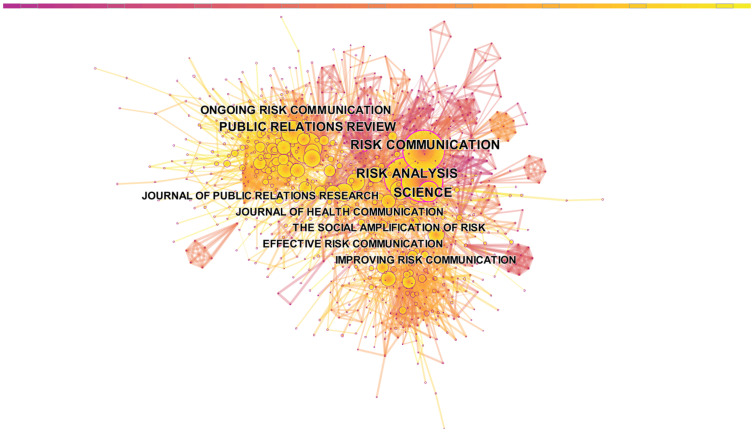
Map of the journal co-citation network of crisis and risk communication research.

**Figure 8 ijerph-19-02923-f008:**
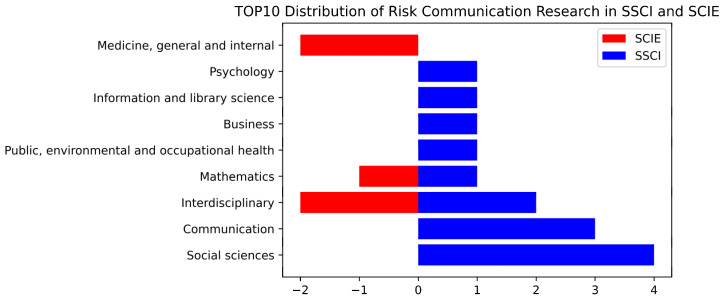
Distribution of top 10 journals crisis and risk communication research in SSCI and SCIE.

**Figure 9 ijerph-19-02923-f009:**
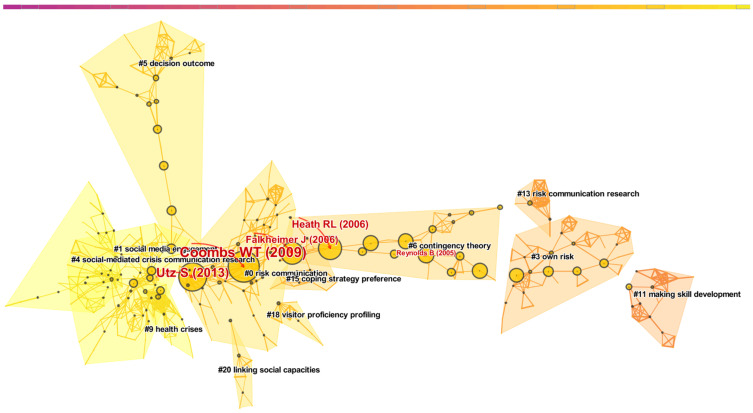
Map of the literature co-citation network of crisis and risk communication research.

**Figure 10 ijerph-19-02923-f010:**
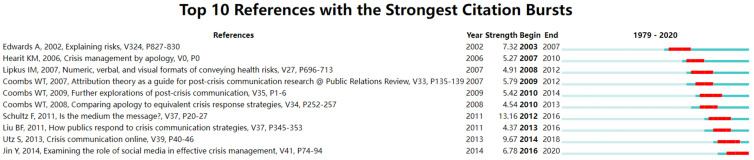
Top 10 references with the strongest citation bursts.

**Figure 11 ijerph-19-02923-f011:**
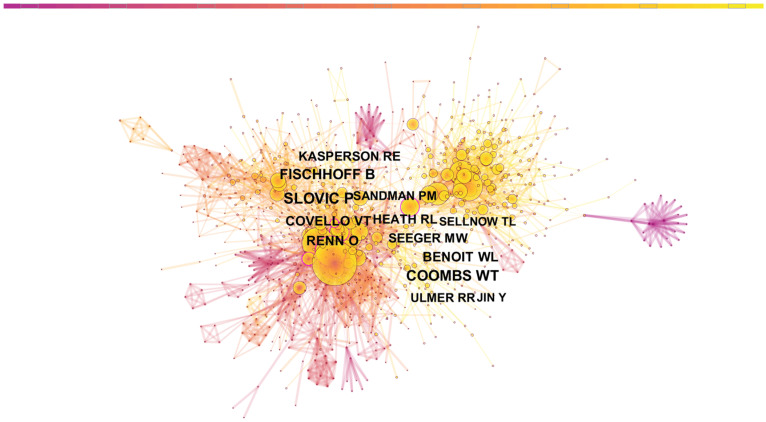
Map of the author co-citation network for crisis and risk communication research.

**Table 1 ijerph-19-02923-t001:** Literature parameters of the combined output from WOS and Scopus.

Total Records	Publications
Article	3679
Book review	58
Correction	15
Editorial material	131
Letter	21
Meeting abstract	175
News item	9
Note	3
Reprint	3
Review	70

**Table 2 ijerph-19-02923-t002:** Node and Network Characteristics.

Figure	Node Threshold	Number of Nodes	Number of Edges	Density
Country co-authorship	100	86	404	0.1105
Institution co-authorship	15	727	1028	0.0039
Author co-authorship	5	1047	2073	0.0038
Journal co-citation	100	1155	3993	0.0060
Literature co-citation	10	1228	6269	0.0083
Author co-citation	100	1130	8008	0.0126
Keywords co-occurrence	100	1121	15,445	0.0246

**Table 3 ijerph-19-02923-t003:** Top 10 countries for crisis and risk communication research.

Ranking ^1^	Counts	Betweenness Centrality	Countries	Year of First Publication
1	1296	0.66	USA	1986
2	271	0.38	England	1989
3	191	0.06	Germany	1989
4	132	0.03	China	1980
5	124	0.16	Netherlands	1994
6	116	0.01	Australia	1991
7	110	0.04	Canada	1995
8	103	0.00	Japan	1997
9	78	0.05	Sweden	2001
10	67	0.00	Italy	1987

^1^ Note. The ranking is based on the number of publications, which are listed in descending order.

**Table 4 ijerph-19-02923-t004:** Top 10 most productive institutions for crisis and risk communication research.

Ranking	Counts	Betweenness	Affiliation	Year of First Publication	Country
1	31	0.04	University of California	1995	USA
2	30	0.01	University of Maryland	2007	USA
3	26	0.03	Wayne State University	1997	USA
4	24	0.01	Rutgers University	1988	USA
5	23	0.03	University of Kentucky	2010	USA
6	23	0.03	Cornell University	1990	USA
7	22	0.02	North Carolina State University	1988	USA
8	22	0.01	Virginia Commonwealth University	2002	USA
9	21	0.03	Harvard University	2002	USA
10	20	0.02	Cardiff University	1999	UK

**Table 5 ijerph-19-02923-t005:** Top 10 most productive authors in crisis and risk communication research.

Ranking	Counts	Betweenness	Author	Year of First Publication	Affiliation
1	19	0.00	Adrian Edwards	1999	Cardiff University, UK
2	16	0.00	W. Timothy Coombs	2007	Texas A&M, USA
3	13	0.00	Ann Fisher	1988	Pennsylvania State University, USA
4	13	0.00	Caron Chess	1989	Rutgers University, USA
5	13	0.00	Robert L. Heath	1995	University of Houston, USA
6	11	0.00	Glyn Elwyn	1999	University of Wales, England
7	10	0.00	Yan Jin	2010	University of Georgia, USA
8	10	0.00	Ortwin Renn	1989	University of Stuttgart, Germany
9	9	0.00	Matthew W. Seeger	2002	Wayne State University, USA
10	9	0.00	Timothy L. Sellnow	1997	University of Central Florida, USA

**Table 6 ijerph-19-02923-t006:** Top 10 most frequently cited journals in crisis and risk communication research.

Ranking	Counts	Betweenness	Cited Journals	Year of First Publication	Category
1	378	0.16	*Risk Analysis*	1997	Social sciences, mathematical methods—SSCI
2	302	0.21	*Science*	1997	Mathematics, interdisciplinary applications—SCIE
3	297	0.05	*Public Relations Review*	1997	Public environmental and occupational health—SSCI
4	133	0.07	*Journal of Health Communication*	1999	Multidisciplinary sciences—SCIE
5	127	0.01	*Journal of Public Relations Research*	2010	Communication—SSCI
6	105	0.04	*Risk Management-An International Journal*	2010	Business—SSCI
7	104	0.03	*Journal of Risk Research*	2005	Communication—SSCI
8	89	0.06	*JAMA Network Open*	2019	Information science and library—SSCI
9	77	0.1	*Journal of Personality and Social Psychology*	1997	Communication—SSCI
10	73	0.03	*BMJ Open*	2012	Social sciences, interdisciplinary—SSCI

**Table 7 ijerph-19-02923-t007:** Main clusters for crisis and risk communication research.

Cluster ID	Size	Silhouette	Label (LLR)	Average Year
0	90	0.934	Risk communication	2008
1	82	0.947	Risk communication	2013
3	47	0.943	Own risk	2003
4	44	0.982	Social-mediated crisis communication research	2016

**Table 9 ijerph-19-02923-t009:** Turning points for crisis and risk communication research.

Paper	Year	Cluster ID
Coombs and Holladay [63]	2009	0
Utz et al. [60]	2013	0
Seeger [70]	2006	6
Falkheimer and Heide [71]	2006	6

**Table 10 ijerph-19-02923-t010:** Top 10 co-cited authors in crisis and risk communication research.

Ranking	Counts	Betweenness	Author	Year of First Publication	Affiliation
1	404	0.08	Paul Slovic	1986	University of Oregon, USA
2	336	0.04	W Timothy Coombs	2001	Texas A&M University, USA
3	247	0.07	Fischhoff B	1988	Carnegie Mellon University, USA
4	201	0.12	Covello VT	1988	Columbia University, USA
5	189	0.03	Benoit WL	1997	University of Alabama at Birmingham, USA
6	185	0.06	Renn O	1994	University of Stuttgart, Germany
7	171	0.02	Seeger MW	2001	Wayne State University, USA
8	149	0.18	Heath RL	1995	University of Houston, USA
9	136	0.10	Kasperson RE	1986	Clark University, USA
10	118	0.09	Sandman PM	1991	Rutgers University, USA

## Data Availability

Not applicable.

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
