# Peer review of "A Visualization Analysis of Crisis and Risk Communication Research Using CiteSpace"

_ijerph, 2022, doi:10.3390/ijerph19052923_

Round 1
Reviewer 1 Report
This paper explores the risk and crisis communication research through use of the software CiteSpace.
Introduction:
- some discussion of the approach and choice of software would be beneficial. Why this type of approach rather than an evidence synthesis approach?
Methods:
- 2.1. I am curious about the search only looking at the title rather than title and abstract. Did the previous studies cited in your introduction also do this?
- 2.1. For your keywords, did you only use 'risk communication' and 'crisis communication'? What about MeSH subject headings, which those are both part of 'Health Communication'? The number of records returned seems very low for this field.
- 2.1 Inclusion criteria- what is your inclusion criteria beyond date? Were all reviews included as well as original articles and letters, etc.?
- 2.2.4. Some description of CiteSpace should be included- what the software does, etc. A citation/reference to the software should be included.
- 2.2.4. What do the various 4 methods for calculating connection strength assess? What do the acronyms stand for?
- 2.3.1. I would define what you mean by co-authorship in terms of country, institution, and author. For example, it is not clear if institution co-authorship means two authors from the same institution or two different institutions.
Results
Interesting visuals included within the results section. The journals, authors, and institutions listed are the ones I would expect within this field, although the overall number of articles coming from many seem very low from what I know of them. Have you looked at their individual research to cross-check that you've captured everything? For example, when you look at Dr. Timothy Sellnow, he has 93 works listed, almost all of which are related to risk and crisis communication https://orcid.org/0000-0001-5353-4629. Perhaps additional keywords or searching within abstracts would help to expand the search and include more relevant research.
Figure 5- why are authors with over 5 citations only shown? Some justification would be helpful.
3.2.2. Begins to get at the information I think would be helpful in making this research more valuable. Dong et al. (2021) and Goerlandt, Li & Reniers (2020) both include a thematic analysis/narrative analysis of focus topics, which is a helpful addition to understand the focus of research in this area. Are you able to include this type of analysis more explicitly? You also have some of this information in 3.3.1 and some expansion here could be helpful.
Discussion
The discussion feels disjointed from the results. Some further integration of research would be helpful here, as well as some discussion on the hot topics and key results from your paper.
Limitations
There should be some inclusion of the limitations associated with your methods. The use of two keywords and two databases likely resulted in missing relevant data. Also, the research seems to be very focused on developed countries.
Reviewer 2 Report
There is no need to include in the abstract subheadings such as methods or results. This is not a structured abstract.
The use of bibliometric maps to represent how different kinds of objects of study (authors, papers, journals, organisations, etc.) are related to one another are considered a useful way to help its visualisation and its comprehension. Many software programs are currently used in order to perform analyses via bibliometric mapping, namely: CiteSpace, VOSviewer, SciMAT, CitNetExplorer, BibExcel, Sci2Tool and Bibliometrix, among others. Each software programme has its advantages and disadvantages. Why CitesSpace was used in this article? Please cite the following articles:
10.3389/fpsyg.2020.629951
10.1002/asi.21525
https://doi.org/10.3390/su1302077
The results obtained does not allow to obtain significant conclusions. For instance, the countries with a higher production are also the countries that create most scientific literature. Is this a relevant contribution to the field? I don’t think so. Likewise, the keyword co-occurrence network is just an automatic process and does not allow the analysts to interpret the results according to their knowledge and expertise. The former methodologies can be used in a paper, but used without a stronger method does not allow to obtain deep conclusions. For this reason, I strongly recommend to include a co-citation analysis of papers in order to detect their clustering, burst papers or trends and turning points. Finally, a research agenda should be included in order to suggest future lines of research once the bibliography has been used (see 10.1111/beer.12400 )
Reviewer 3 Report
The authors made a study during their research to understand the research status and development trend of Crisis and Risk Communications Research (CRCR) through visual analysis in CiteSpace, thereby providing multinational companies and researchers with a broader perspective to understand CRCR. Their modus operandi is that they first retrieved published articles from the Web of Science (1986-2020) and Scopus (1979-2020) using a title search. Furthermore, they analyzed the main research strength and topic development of CRCR from three dimensions such as co-authorship network, co-citation network and keyword co-occurrence network. The work is very interesting and certainly useful for the bibliometrics community. But there are a few explanations missing in various places and, above all, a state of the art, there are different authors who have also carried out the same analysis. You can look it up in the literature and compare your solution. What makes your study different from the others? This potential of the solution still has to be processed and clarified in the work. Because it can't be, only you have covered this topic so far.
Round 2
Reviewer 1 Report
Thank you for your response and the integration of my suggestions into the revised manuscript. The changes add important detail that makes the findings easier to interpret.
Author Response
Thank you again sir~
Reviewer 2 Report
The authors improved the document but there is room for improvement in regard to some results
